# Inhibition of NHEJ repair by type II-A CRISPR-Cas systems in bacteria

Aude Bernheim[1,2,3,4], Alicia Calvo-Villamañán[3], Clovis Basier[3], Lun Cui [3], Eduardo P.C. Rocha[1,2], Marie Touchon[1,2] & David Bikard [3]

Type II CRISPR-Cas systems introduce double-strand breaks into DNA of invading genetic material and use DNA fragments to acquire novel spacers during adaptation. These breaks can be the substrate of several DNA repair pathways, paving the way for interactions. We report that non-homologous end-joining (NHEJ) and type II-A CRISPR-Cas systems only co-occur once among 5563 fully sequenced prokaryotic genomes. We investigated experimentally the possible molecular interactions using the NHEJ pathway from *Bacillus subtilis* and the type II-A CRISPR-Cas systems from *Streptococcus thermophilus* and *Streptococcus pyogenes*. Our results suggest that the NHEJ system has no effect on CRISPR immunity. On the other hand, we provide evidence for the inhibition of NHEJ repair by the Csn2 protein. Our findings give insights on the complex interactions between CRISPR-Cas systems and repair mechanisms in bacteria, contributing to explain the scattered distribution of CRISPR-Cas systems in bacterial genome.

[1] Microbial Evolutionary Genomics, Institut Pasteur, 25-28 rue Dr. Roux, 75015 Paris, France. [2] CNRS, UMR3525, 25-28 rue Dr. Roux, 75015 Paris, France. [3] Synthetic Biology Group, Institut Pasteur, 25-28 rue Dr. Roux, 75015 Paris, France. [4] AgroParisTech, 75005 Paris, France. Marie Touchon and David Bikard contributed equally to this work. Correspondence and requests for materials should be addressed to D.B. (email: david.bikard@pasteur.fr)

CRISPR (Clustered Regularly Interspaced Short Palindromic Repeats) arrays and their associated (Cas) proteins confer Bacteria and Archaea adaptive immunity against phages and other exogenous mobile genetic elements[1–4]. Yet, although most bacteria are infected by phages and other mobile genetic elements, CRISPR-Cas systems are absent from the majority of bacterial genomes[5,6]. The selective pressures and mechanisms that lead to the success of CRISPR-Cas systems in some clades and not others remain poorly understood.

CRISPR-Cas systems are classified into 6 types and 27 subtypes, according to the Cas proteins they carry[5,7]. The recent development of CRISPR-Cas9-based genetic engineering technologies has made type II CRISPR-Cas systems the focus of many investigations. Type II systems include the CRISPR array, three core genes (*cas1*, *cas2*, and *cas9*), and a small *trans*-activating CRISPR RNA (crRNA) complementary to the CRISPR repeat sequence[8,9]. A fourth gene is involved in spacer acquisition, *csn2* in the type II-A[4,10,11], and *cas4* in type II-B systems[8]. A third subtype, type II-C, only requires *cas1*, *cas2*, and *cas9*[5,8]. All the Cas proteins of type II systems are necessary for spacer acquisition[10,11], but only Cas9 is necessary for interference[12,13]. The Cas9 protein is guided by small crRNA to introduce double-strand breaks (DSB) into target DNA[12,14]. A short conserved sequence (2–5 bp) adjacent to the protospacer known as the PAM (protospacer adjacent motif) is essential to distinguish foreign from self DNA and can be different for CRISPR-Cas systems of the same type[15,16].

In bacteria, DSB can be repaired either by homologous recombination (HR) or by non-homologous end-joining (NHEJ). These mechanisms could thus affect the efficiency of CRISPR-Cas interference by repairing the breaks. Type II CRISPR-Cas systems introduce DSB at the same position in all copies of the target DNA molecule[17], and the concomitant lack of an intact DNA template should preclude the repair of these DSB by HR. However, NHEJ repairs DSB without requiring template DNA[18] and could mend DSB generated by Cas9. In Eukaryotic cells, breaks introduced by Cas9 can efficiently be repaired by NHEJ, a strategy now widely used to introduce indel mutations[19]. In bacteria, the NHEJ system requires two core proteins: Ku and a ligase[20]. Ligation is usually carried out by the LigD protein, but other ligases can be recruited by Ku when LigD is absent[18]. The system is complemented by additional proteins in certain cases[21]. Ku binds at the DSB and recruits the ligase to seal the break[22,23]. NHEJ offers a mean to repair DSB when only a single copy of the genome is available, such as after sporulation or during stationary phase[24,25]. NHEJ repair can be mutagenic[26], leading to up to 50% error rates in certain bacteria[23].

DNA repair pathways could also affect the acquisition of novel spacers by CRISPR-Cas systems because they modulate the availability of DSB and/or compete with the Cas machinery for the DNA substrate. Conversely, the action of Cas proteins at DSB could hinder DNA repair pathways. It was show`n that novel spacers of type I CRISPR-Cas systems can be acquired after DSB from RecBCD degradation products[27]. Importantly, DNA repair pathways and CRISPR-Cas systems are composed of proteins with structural similarities and interacting with the same substrates[28]. For example, Cas4, a protein present in type I and type II-B systems shares structural and functional similarities with AddB[28,29], a component of the AddAB repair pathway and a functional homolog of RecBCD[30]. In type II-A CRISPR-Cas systems, Csn2 binds and slides along free DNA ends in the same manner as the Ku protein of the NHEJ system[28,31–34]. If Cas proteins and proteins involved in DNA repair mechanisms recognize the same substrate, a competition might arise leading to antagonistic interactions between the two processes.

The interaction between the NHEJ system and Cas9 is at the heart of the CRISPR-Cas-based genetic engineering technologies, and we now investigate it in bacteria. We hypothesize that the NHEJ system could interfere with the activities of type II CRISPR-Cas systems by repairing DSB generated by Cas9 during interference or by competing with Cas proteins for the same substrate during adaptation. Alternatively, type II CRISPR-Cas systems could interfere with NHEJ during repair. We test these hypotheses by assessing the patterns of co-occurrence of the two systems in bacterial genomes. This reveals one single case of co-occurrence of both systems among 5563 bacterial genomes, suggesting strong negative interaction. We then investigate experimentally mechanisms that could explain this interaction, by introducing the NHEJ system from *B. subtilis* and/or the CRISPR-Cas system from *S. pyogenes* in *B. subtilis*, *S. thermophilus*, and *S. aureus*. We could not measure any effect of the NHEJ system on type II-A CRISPR-Cas interference and adaptation. On the other hand, our results suggest that the Csn2 protein inhibits NHEJ repair.

## Results

**Negative association between NHEJ and type II-A CRISPR-Cas.** We detected CRISPR-Cas and NHEJ systems in 5563 fully sequenced bacterial genomes (Supplementary Data 1). The NHEJ pathway was present in 24.7% and the type II CRISPR-Cas system in 6.9% of the genomes, and these systems were very unevenly distributed among bacterial phyla (Supplementary Fig. 1 and Supplementary Table 1). Firmicutes and Proteobacteria were the only phyla with genomes encoding enough type II CRISPR-Cas systems (respectively 209 and 101) and NHEJ (respectively 364 and 637), to perform robust statistical analyses (Supplementary Fig. 1). A possible confounding factor when studying the distribution of bacterial defense and DNA repair pathways is that their abundance co-vary with genome size[35,36]. Accordingly, NHEJ systems were more frequent in larger genomes ($P < 10^{-4}$, $\chi^2$ test on a logistic fit). In contrast, type II CRISPR-Cas systems were only present in genomes smaller than 5 Mb (Supplementary Fig. 2). Hence, we focused our analysis on Firmicutes and Proteobacteria with genomes smaller than 5 Mb. They represent 56.5% of the total number of genomes. In this sample, the size of the genomes encoding NHEJ systems was independent of the presence of a type II CRISPR-Cas system ($P = 0.99$, Wilcoxon's test).

We analyzed the patterns of co-occurrence of NHEJ and CRISPR-Cas systems to test if they were independently distributed. We observed that NHEJ and type II systems were negatively associated in Firmicutes ($P < 10^{-4}$, Fisher's exact test), but not in Proteobacteria ($P = 0.70$, Fisher's exact test) (Fig. 1b and Supplementary Fig. 3). Note however that different subtypes of type II CRISPR-Cas systems are distributed differently in these two phyla. Proteobacteria encoded many type II-C and no type II-A systems, whereas Firmicutes encoded mostly type II-A systems (Fig. 1a). Type II-B systems were only detected in nine genomes and thus were not further analyzed. To test if different subtypes could have different interactions with NHEJ systems, we looked at them separately. When studying co-occurrences of genes, it is important to consider that genomes are linked by a common evolutionary history, which decreases the degrees of the freedom of the statistical analyses. To check whether systems are negatively associated while taking phylogeny into account, we built a tree of Firmicutes and tested if the binary traits (presence of both systems) evolved independently using BayesTraits[37]. A strong negative association between NHEJ and type II-A CRISPR-Cas systems was observed (Bayes factor (BF) = 9.7, Fig. 1c), while no associations between NHEJ and type II-C

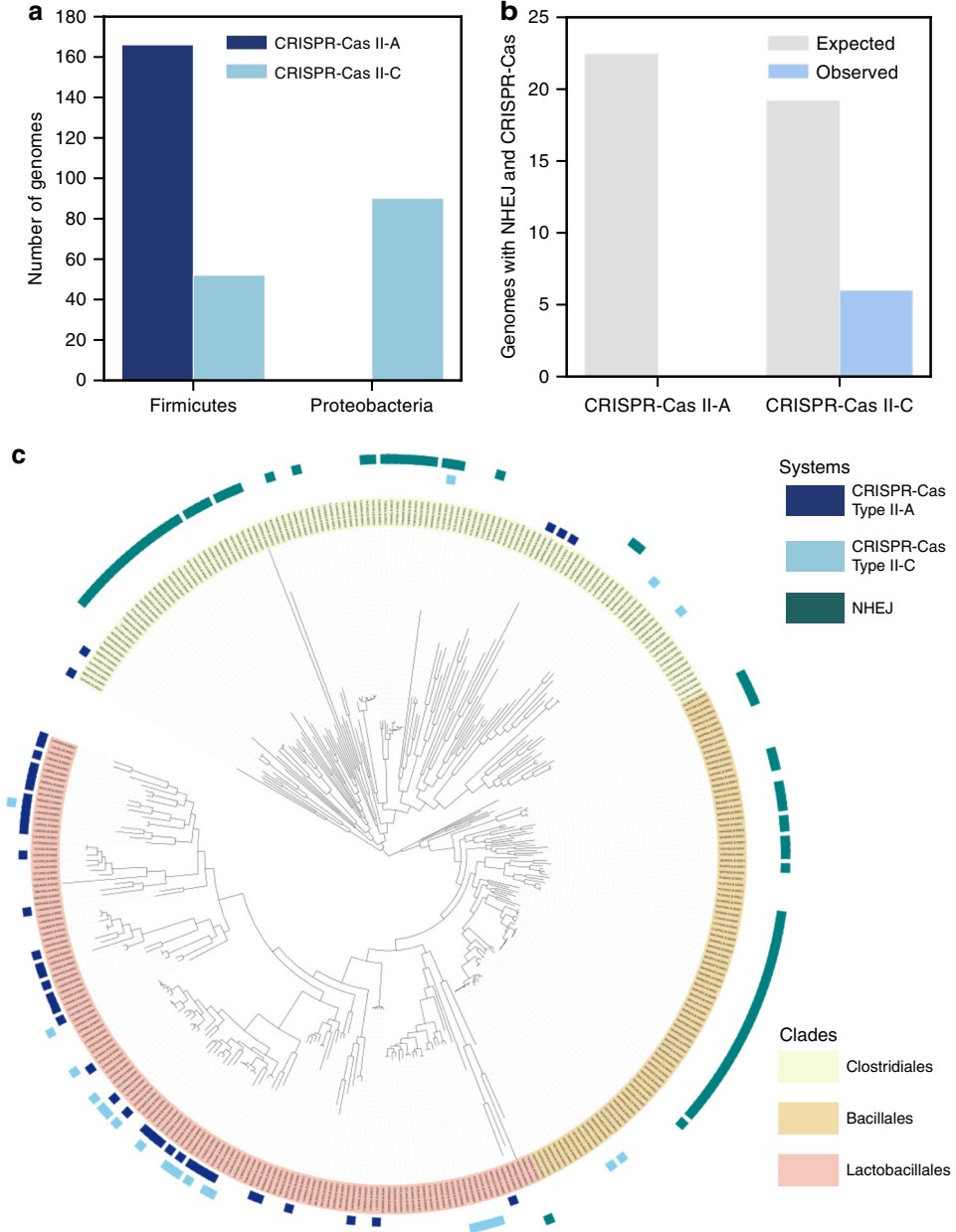

**Fig. 1** Negative association between NHEJ and type II-A CRISPR-Cas systems. **a** Distribution of the subtypes II-A and II-C in Proteobacteria and Firmicutes genomes. **b** Associations between NHEJ and subtypes II-A and II-C CRISPR-Cas systems. Expected values correspond to the number of co-occurrences that would be obtained if the systems were randomly distributed. **c** Presence of NHEJ and type II CRISPR-Cas systems in Firmicutes. A system is annotated as present in a given species when more than half of the genomes available for this species encode the system

CRISPR-Cas systems was detected. Only one genome among the 5563 encodes both NHEJ and type II-A: the actinobacteria *Eggerthella* sp. *YY7918*. In this genome, both NHEJ and type II-A systems seem intact, since the *cas* operon contains all four genes, with no frameshifts or premature stop codons, and the adjacent CRISPR array encodes 44 spacers. We were also unable to detect in this *Eggerthella* genome anti-CRISPR proteins similar to the ones described in the literature[38–40].

**NHEJ has no measurable effect on type II-A CRISPR-Cas interference**. We first tested if the *B. subtilis* NHEJ system could affect type II-A CRISPR-Cas interference, using the previously described *S. aureus* model system[10]. The *ku* and *ligD* genes were cloned under the control of a Ptet promoter (plasmid pAB1) into

*S. aureus* RN4220 cells. This system was able to circularize linearized plasmids after electroporation, showing it is functional (Supplementary Note 1 and Supplementary Fig. 4). The type II-A CRISPR-Cas system from *S. pyogenes* was introduced on plasmid pDB114 and programmed with a single spacer targeting phage phiNM4 (pMD021). *Streptococcus aureus* cells carrying both systems were then challenged in phage infection assays. A NHEJ system might facilitate phage escape from CRISPR-Cas by promoting the introduction of mutations at the target site through unfaithful repair, or by efficiently and faithful repairing DSB generated by Cas9, making CRISPR immunity inefficient.

First, the unfaithful repair of Cas9 breaks could lead to the formation of indels that would block further cleavages. The generation of such mutant phages should lead to a higher efficiency of plaquing (EOP) of phiNM4 when the NHEJ system

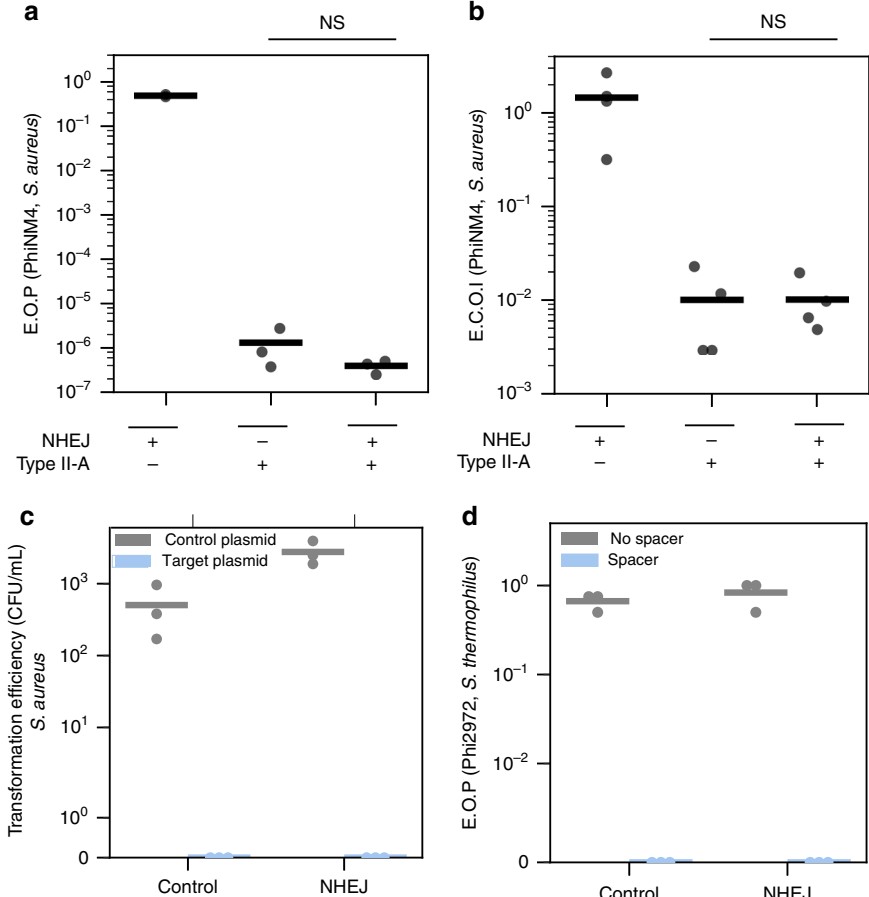

**Fig. 2** The *B. subtilis* NHEJ system has no measurable effect on type II-A CRISPR-Cas interference. **a** Resistance to phage phiNM4 provided by the *S. pyogenes* CRISPR-Cas9 system in *S. aureus* in the presence (pAB1) or absence (pE194) of the NHEJ system from *B. subtilis* ($n = 3$, mean, NS double-sided $t$ test $P = 0.9999$). **b** Efficiency of center of infection (ECOI), i.e., the proportion of cells that produce at least one functional phage particle, in the presence (pAB1) or absence (pE194) of the NHEJ system ($n = 4$, mean, NS double-sided $t$ test $P = 0.9998$). **c** Transformation efficiency of plasmid pT181 either empty or carrying a target sequence (pAB2) in *S. aureus* RN4220 cells expressing the CRISPR-Cas system from plasmid pMD021 in the presence (pAB1) or absence (pE194) of the NHEJ system from *B. subtilis* ($n = 3$, mean). **d** EOP of phage Phi2972 on a bacteriophage-insensitive mutant of *S. thermophilus* DGCC7710 carrying a spacer against Phi2972. Cells express either the *B. subtilis* NHEJ system from plasmid pAB66 or a control GFP from plasmid pAB69 ($n = 3$, mean)

is expressed. The CRISPR-Cas system provided a five order of magnitude reduction in the EOP of phage phiNM4 when compared with a spacer-less control, and no significant increase in the number of plaques was observed upon NHEJ induction (Fig. 2a). To confirm that the small number of plaques obtained could not result from the unfaithful repair of Cas9 breaks through NHEJ, we sequenced the target position of eight mutant phages. All mutants had a point mutation in the PAM and none presented an indel.

Second, the faithful repair of Cas9 breaks could lead to a cycle of repair and cleavage that would allow the production of functional phage particles. In this case, it might not be possible to observe plaque formation as the competition between NHEJ and CRISPR interference might lower burst sizes. To test this hypothesis, we measured the efficiency of center of infection (ECOI), i.e., the number of cells that produce at least one functional phage particle after infection compared to the control strain (sensitive to the phage). One would expect higher ECOI of phiNM4 when cells express the NHEJ system. The observed ECOI was ~$10^{-2}$ regardless whether the NHEJ system was induced or not (Fig. 2b).

We further tested whether NHEJ could reduce CRISPR-Cas9 immunity against plasmids. To this end, we cloned the PhiNM4 target sequence used above on plasmid pAB2 and transformed

this plasmid in strains carrying the NHEJ system or not. While a control target-less plasmid could be efficiently introduced in the cells, no clones were recovered after transformation of pAB2 regardless of the presence of the NHEJ machinery. This shows that the CRISPR-Cas system efficiently blocks plasmid transformation and that the NHEJ system did not measurably reduce the efficiency of CRISPR immunity, nor introduced mutations in the target plasmid at a detectable rate (Fig. 2c).

To confirm these results in a bacterium that naturally carries a type II-A CRISPR-Cas system, we measured interference against phage Phi2972 in *S. thermophilus*, in the presence or absence of the NHEJ system from *B. subtilis*. Genes *ku* and *ligD* were cloned under the control of a constitutive promoter on plasmid pNZ123 and introduced in a derivative of strain DGCC7710 whose CRISPR-1 locus carries a spacer targeting phage Phi2972. The resistance provided by the CRISPR-Cas system was as strong in the presence of the NHEJ system as in the presence of a control GFP carried by the same plasmid (Fig. 2d). All in all, our results do not support the hypothesis that NHEJ affects type II-A CRISPR-Cas interference.

**NHEJ has no measurable effect on type II-A CRISPR-Cas acquisition.** Ku and Csn2 bind the same type of substrate—linear

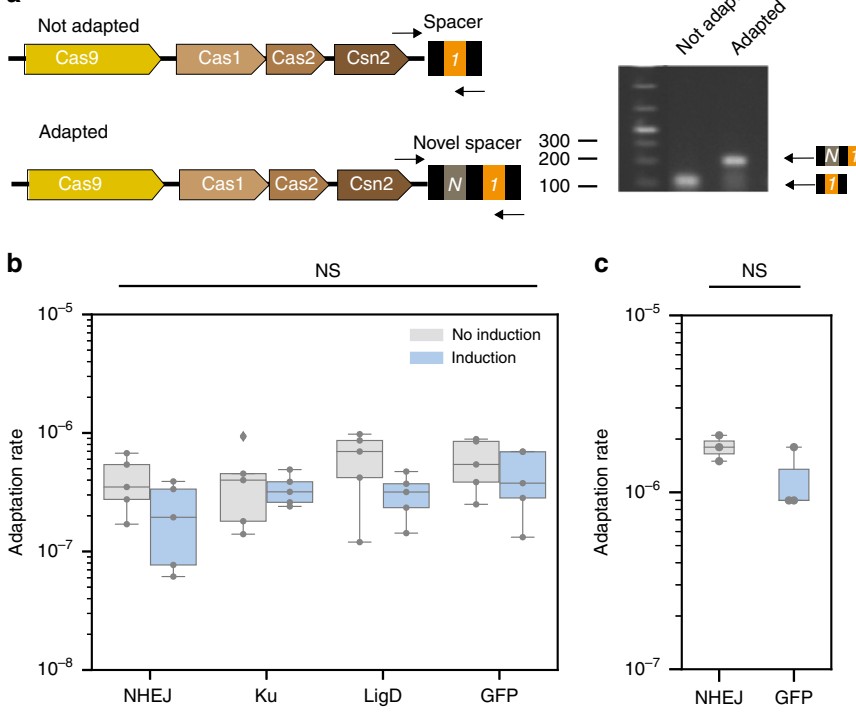

**Fig. 3** The *B. subtilis* NHEJ system has no measurable effect on spacer acquisition. **a** *S. aureus* strain RN4220 carrying the type II-A CRISPR-Cas system on plasmid pRH87 was challenged with phage phiNM4. Spacer acquisition was assessed by PCR on isolated colonies that survived the infection (oligonucleotides are depicted as black arrows). **b** Adaptation rate measured in the presence of NHEJ, ku, ligD, or GFP carried by plasmids pAB23, pAB24, pAB25, and pAB62, respectively (*n* = 5, ANOVA, NS *P* = 0.5674). **c** Adaptation rate of *S. thermophilus* DGCC7710 against phage Phi2972 when expressing the *B. subtilis* NHEJ system from plasmid pAB66 or a control GFP from plasmid pAB69 (*n* = 3, two-sided *t* test, NS *P* = 0.91)

double-stranded DNA[28,33]—and might thus interfere antagonistically. To test if the NHEJ system affects spacer acquisition, we measured the cells' ability to acquire new spacers in the presence of the NHEJ machinery. *Streptococcus aureus* cells carrying the NHEJ system (pAB1) and the type II-A CRISPR-Cas system (pRH87) were infected by phage PhiNM4 either with or without induction of the NHEJ system[10]. In this experiment, cells can escape phage infection either by capturing a novel spacer or by using other mechanisms of defense. Survivors were screened by PCR to check for acquisition of novel spacers and measure adaptation rate (Fig. 3a). No effect of the NHEJ system on the adaptation rate was observed. As a control, the expression of Ku alone, ligD alone, or GFP were also observed to have no effect (analysis of variance, *P* = 0.16) (Fig. 3b).

To corroborate these results, a similar experiment was performed in *S. thermophilus*. Cells carrying the *B. subtilis* NHEJ system or a control GFP on a plasmid were infected with phage Phi2972. We observed no difference in the rate of novel spacer acquisition between cells expressing the NHEJ machinery or the GFP (Wilcoxon's test, *P* = 0.26) (Fig. 3c). Altogether, these results indicate that NHEJ has no effect on the acquisition of novel spacers by a type II-A CRISPR-Cas system.

**Csn2 inhibits NHEJ repair**. As Csn2 binds to the same substrate as Ku, it could interfere with NHEJ repair[28,33]. To test this hypothesis, we reproduced the experiment that led to the discovery of the NHEJ system in *B. subtilis*[41]. When *B. subtilis* cells in stationary phase are irradiated by ionizing radiations, the DSB generated are repaired by the NHEJ system, as other repair systems cannot function in those specific conditions. *Bacillus subtilis* deleted for NHEJ do not survive irradiation as well as the wild

type. If type II-A CRISPR-Cas systems limit NHEJ repair, cells bearing a type II-A CRISPR-Cas system are expected to show increased sensitivity to irradiation.

*B. subtilis* cells expressing the type II-A CRISPR-Cas system from plasmid pRH087 were more sensitive to irradiation than cells carrying a control empty vector and showed the same level of sensitivity as the Δku-ligD mutant (*P* < 10⁻⁴, Wilcoxon's test, Fig. 4a). If the increased sensitivity provided by the CRISPR-Cas system is due to an impairment of NHEJ repair, then we expect to observe no cumulative effects when the NHEJ system is deleted and the CRISPR-Cas system added. Indeed, cells deleted for the NHEJ system and carrying the type II-A CRISPR-Cas system have the same survival as the ones deleted for the NHEJ system, pointing towards an interaction between the two systems. Another prediction that results from this hypothesis is that the CRISPR-Cas system should have no effect on the sensitivity to irradiation in species that lack a NHEJ system. To test this, we performed irradiation experiments on *S. aureus* cells carrying plasmid pRH87 or the control pC194. The presence or absence of the CRISPR-Cas system did not have an effect on survival in *S. aureus* (*P* = 0.5, Wilcoxon's test, Supplementary Fig. 5). Taken together, these results support the hypothesis that the type II-A CRISPR-Cas system impairs the NHEJ system.

To understand if a specific protein was responsible for this phenotype, we deleted or mutated individual *cas* genes from plasmid pRH87 and performed the same assay. While the effect size is small, the only mutant that significantly rescued *B. subtilis* cells upon irradiation was the delta *csn2* mutant (*P* = 0.02, Student's two-sided *t* test after validation of normality and homoscedasticity, Fig. 4b). When expressed alone, Csn2 was able to decrease survival of irradiated cells to the same level as the whole CRISPR-Cas system, while no effect could be observed

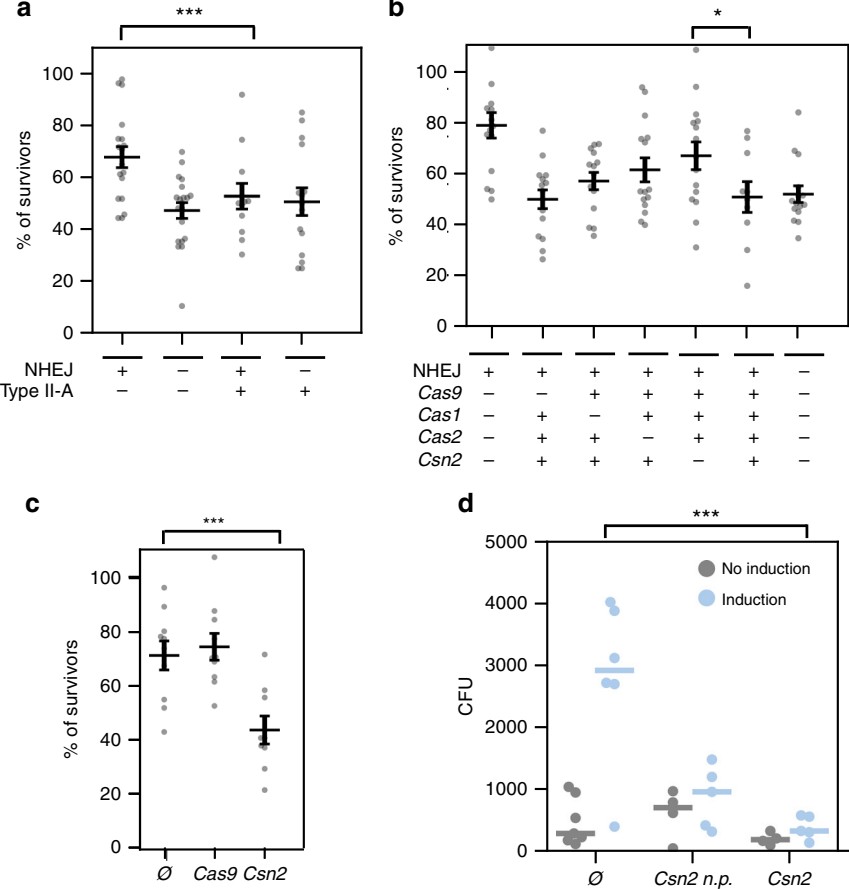

**Fig. 4** Csn2 impacts NHEJ repair. Survival rates of irradiated *B. subtilis* cells. **a–c** Individual replicates (points) and average (horizontal bars) are shown. Error bars correspond to the standard error of the mean (s.e.m.). **a** Cells carrying the type II-A CRISPR-Cas system (pRH87) or the control empty vector (pC194), and deleted for *ku* and *ligD* or not ($P = 0.0009$, Wilcoxon's). **b** *B. subtilis* carrying the CRISPR-Cas system with the dCas9 mutations (pRH121) or deleted for *csn2* (pRH63), *cas1* (pRH61), or *cas2* (pRH62) ($P = 0.02$, Student's two-sided *t* test after validation of normality and homoscedasticity). **c** *B. subtilis* carrying the empty pC194 plasmid (Ø), expressing *csn2* from plasmid pAB56 or *cas9* from plasmid pDB114 ($P = 0.0048$, Wilcoxon's). **d** A linearized plasmid providing resistance to chloramphenicol (pC194) was electroporated into *S. aureus* RN4220 cells carrying the NHEJ system either alone (plasmid pAB1, Ø) or with *csn2* cloned downstream of *ligD* (plasmid pAB81, csn2) or under the control of its natural promoter (plasmid pAB82, csn2 n.p.). The number of CFUs obtained with or without induction of the NHEJ system using aTc are reported. The number of CFU obtained without induction (gray bars) indicate the background of already circular DNA present in the sample before electroporation ($P = 0.006$, two-sided *t* test)

with an empty vector or Cas9 alone ($P < 10^{-4}$, Wilcoxon's test, Fig. 4c). In this set of experiments a possible concern is that Csn2 might be overexpressed which could lead to artifacts with no biological relevance. To prevent this issue, we expressed the whole *S. pyogenes* type II-A system or Csn2 alone from the natural promoter of the *cas* operon (plasmid pRH87 and pAB56, respectively). The expression of Csn2 in *B. subtilis* as measured by quantitative polymerase chain reaction (qPCR) was 3.6-fold lower than the basal expression level of Csn2 in *S. pyogenes* SF370 (Supplementary Fig. 6). This low level of expression might reflect what would happen after a natural horizontal gene transfer event.

To obtain more direct evidence that Csn2 blocks NHEJ repair, we investigated its ability to inhibit the recircularization of linear plasmid DNA upon electroporation into *S. aureus*. The *csn2* gene was added to plasmid pAB1 which encodes Ku and LigD, either under the control of a Ptet promoter (pAB82), or under the control of the *cas* operon promoter (pAB81). We then electroporated a linearized plasmid providing resistance to chloramphenicol (pC194) into cells expressing the NHEJ system or both NHEJ and Csn2 (protocol presented in Supplementary Fig. 4.a). The *B. subtilis* Ku and LigD were able to circularize the plasmid DNA in *S. aureus*, but we obtained on average fivefold fewer colonies when Csn2 was co-expressed with Ku and LigD

compared to the NHEJ system alone (Fig. 4d). In this assay the NHEJ system is strongly overexpressed compared to the natural expression of Ku and LigD in *B. subtilis* during stationary phase (Supplementary Fig. 6). Note that such overexpression was necessary to observe plasmid recircularization events in *S. aureus*. On the other hand, Csn2 was only slightly overexpressed compared to its expression level in *S. pyogenes* SF370. Altogether, these results show that Csn2 hinders NHEJ repair.

## Discussion

We found that with the exception of a single case, NHEJ and type II-A CRISPR-Cas systems do not co-occur in fully sequenced bacterial genomes available to date. A possible incompatibility between NHEJ and type II-A CRISPR-Cas systems was investigated in a variety of experimental systems encompassing *S. aureus*, *B. subtillis*, and *S. thermophilus*. Our results indicate that NHEJ does not affect CRISPR immunity against phages and plasmids, nor the capture of novel spacers. Previous studies showed that NHEJ repair pathways are able to repair Cas9-mediated DNA breaks in various bacterial species[17,42]. The efficiency of repair in these experimental setups was very low. Consistently, our results show that NHEJ repair cannot lead to a

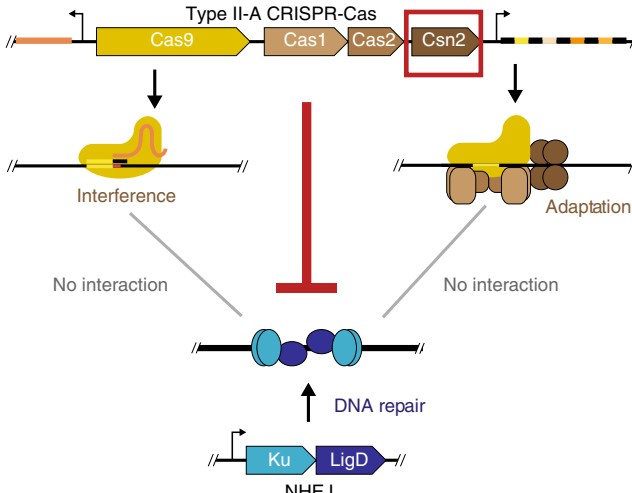

**Fig. 5** Graphical summary of the results. Three possible modes of negative interactions between type II-A CRISPR-Cas systems and NHEJ systems were tested: NHEJ could block CRISPR interference, NHEJ could block CRIPSR adaptation, or CRISPR could block NHEJ repair. The last hypothesis was shown to be correct and Csn2 to be responsible for the inhibition of NHEJ repair

meaningful reduction in phage infectivity or plasmid transfer. Our results rather show that the Csn2 protein from type II-A CRISPR-Cas systems is able to inhibit NHEJ repair (Fig. 5).

The strong avoidance of co-occurrences between NHEJ and type II-A systems was not observed with type II-C systems. This is consistent with the fact that type II-C systems lack Csn2. Csn2 is a multimeric toroidal protein that can bind double-stranded DNA ends and slide inward through rotation-coupled translocation[28]. These DNA binding properties were noted in previous reports to be very similar to that of the Ku protein[28]. When present in the same cell, these two proteins will likely compete for the same substrate. We suggest that the binding of Csn2 at DNA ends could block access to Ku or inhibit its function preventing efficient repair by the NHEJ machinery.

CRISPR-Cas systems are present in 47% of fully sequenced bacterial genomes[5] and this frequency might be much smaller in uncultivated bacteria[43]. This is in striking contrast with other defense systems, such as restriction modification systems, present on average at two copies per genome[35]. CRISPR-Cas systems are known to be transferred horizontally at a high rate[44], suggesting that they should spread in the bacterial world very rapidly if they were always advantageous. This brings to the fore the intriguing question of what is preventing further CRISPR rise in bacteria. Hypotheses that have been put forward include the cost of autoimmunity, the cost of limiting horizontal gene transfer, and the cost of inducible defenses[45–48]. Our results suggest another (non-mutually exclusive) reason: negative epistasis between the genetic background of a bacterium and a CRISPR-Cas system acquired by horizontal transfer can lead to a decreased fitness. In the present case, the type II-A CRISPR-Cas system affects the efficiency of NHEJ repair, thereby decreasing the fitness gain associated with the acquisition of the system. Note that type II-A systems are constitutively expressed in the bacteria where they have been studied (S. pyogenes[9], S. thermophilus[49]), and would thus likely also be expressed in the recipient upon horizontal gene transfer. We therefore propose that reliance on NHEJ repair is a barrier to the establishment of this type of CRISPR-Cas systems in bacteria.

We have observed an intriguing tendency of type II CRISPR-Cas systems to be absent from the largest genomes. DNA repair

mechanisms are more frequent in larger genomes, presumably as a result of the presence of more abundant accessory functions[50], and to maintain constant genomic mutation rates[51]. If these larger genomes endure stronger selection for the presence of NHEJ, then incoming type II-A CRISPR-Cas systems will not be maintained in the genome. In agreement with the hypothesis of a trade-off between the two functions, nearly all of the largest genomes of Firmicutes encode NHEJ systems.

Sorek and colleagues[27] previously reported a positive effect of recBCD function on type I-E CRISPR spacer acquisition. Since CRISPR-Cas systems acts by cutting DNA, interactions between these systems and DNA repair pathways might be numerous. These interactions are not only relevant to the evolution of bacterial genomes, but are also at the core of CRISPR genome editing technologies which rely on the repair of DNA breaks generated by Cas nucleases. In particular, the ability of Csn2 to block NHEJ repair could prove especially useful in genome editing experiments performed in Eukaryotes where NHEJ repair of Cas9-mediated breaks can compete with homology-directed repair and limit the efficiency of precise editing.

## Methods

**Detection of repair systems and CRISPR-Cas systems**. NHEJ and type II CRISPR-Cas systems were detected using MacSyFinder (default parameters)[6]. The published models were used for the detection of type II CRISPR-Cas systems[6]. To detect NHEJ, we retrieved protein profiles from TIGRFAM: Ku (PF02735) and ligD (TIGR02777, TIGR02778, TIGR02779). We built a MacSyFinder model where the presence of Ku was defined as mandatory and that of LigD as accessory (Supplementary Note 2). Other ligases can indeed be recruited by Ku[18]. With this method, 74% of the systems detected encoded both Ku and LigD; 26% encoded only Ku. We compared these results to a previous analysis using other methods[30]. Only one out of 113 genomes was discordant (we identified a NHEJ system in *Sinorhizobium meliloti* were none had been found previously)[30].

**Genome dataset**. We analyzed 5563 complete genomes retrieved from NCBI RefSeq (ftp://ftp.ncbi.nih.gov/genomes/, last accessed in November 2016) representing 2437 species of Bacteria. These can be found in Supplementary Data 1.

**Phylogenetic analyses**. We built persistent genomes for 245 Firmicutes genomes smaller than 5 Mb available in GenBank RefSeq (Dataset). The persistent genome of each clade was defined as the intersection of pairwise lists of orthologs that were present in at least 90% of the genomes. A list of orthologs was identified as reciprocal best hits using end-gap free global alignment, between the proteome of a pivot and each of the other strain's proteomes. *Bacillus subtilis* strain 168 was used as pivot for each clade. Hits with <37% similarity in amino acid sequence and more than 20% difference in protein length were discarded. We made a persistent genome tree by concatenation of the multiple alignments of the persistent genes obtained with MAFFT v.7.205 (with default options, PMID: 23329690) and BMGE (with default options, PMID: 20626897). Missing genes were replaced by stretches of "-" in each multiple alignment. The tree was computed with IQ-TREE multicore v.1.5.4 under the LG + R10 model[52]. This model gave the lowest Bayesian Information Criterion (BIC) among all models available (option −m TEST in IQ-TREE). We made 1000 ultra-fast bootstraps to evaluate node support (options −bb 1000 −wbtl in IQ-TREE).

We applied BayesTraits v.2.0[37] to test the correlations among pairs of traits that adopt a finite number of discrete states. We ran two models (Independent and Dependent) in MCMC mode (priorAll exp 10) and computed the BF which can be interpreted as follows: <2 weak evidence, >2 positive evidence, 5–10 strong evidence, and >10 very strong evidence[53].

**Bacterial strains and growth conditions**. *S. aureus* strain RN4220 was grown in TSB (Tryptic Soy Broth) or TSA (Tryptic Soy Agar) at 37 °C. Whenever applicable, media were supplemented with chloramphenicol (10 μg ml⁻¹), erythromycin (10 μg ml⁻¹), tetracycline (100 ng mL⁻¹), or spectinomycine (120 μg ml⁻¹) to ensure the maintenance of pC194-derived, pE194-derived, pT181-derived, and pLZ-derived plasmid, respectively. Expression from ptet promoters was induced by the addition of anhydrotetracycline (aTc) at 0.5 μg mL⁻¹. *Streptococcus thermophilus* strain DGCC7710 was grown in LM17 at 37 °C. Whenever applicable, media were supplemented with chloramphenicol (5 μg ml⁻¹) to ensure the maintenance of pNZ123-derived plasmids. *Bacillus subtilis* strain 168 was grown in LB or LB agar at 37 °C. Whenever applicable, media were supplemented with chloramphenicol (5 μg ml⁻¹) or erythromycin (1 μg ml⁻¹) to ensure the maintenance of pC194-derived plasmids and the integration of pMUTIN4-derived plasmids.

**Plasmids and strains construction**. The cloning strategies employed for each plasmid are summarized in Supplementary Table 2 and the primers used are listed in Supplementary Table 3. PCR fragments were assembled using Gibson assembly[54] unless mentioned otherwise. Plasmids pAB2, pAB17, pAB18, and pAB56 were obtained by PCR followed by blunt end ligation. Plasmid pMD021 was assembled by Golden Gate[55].

**NHEJ functionality assay**. The plasmid pC194 was linearized by PCR using primers B329 and B330 (Supplementary Table 3). Strains with the plasmids carrying the NHEJ system were grown to optical density (OD) 0.3 and the NHEJ system was induced by adding aTc. Cells were grown to OD 0.8 and made electro-competent by washing them three times in ice-cold water, supplemented with 10% glycerol for the last wash, and concentrated 100-fold. We transformed 200 μg of linearized pC194 in those electro-competent cells and added aTc to the recovery medium. Cells were plated on selective media and incubated overnight at 37 °C. We resuspended single colonies in lysis buffer with 15 ng mL$^{-1}$ lysostaphin and incubated them at 37 °C for 10 min, then 98 °C for 10 min. Following centrifugation (11 000 g), 1 μL of the supernatant was used as template for DreamTaqPCR amplification with primer A9, A10 (Supplementary Table 3). PCR products were then purified and sequenced.

**CRISPR-Cas interference efficiency assay using phages**. We used two types of assays to assess the impact of Ku and LigD on CRISPR-Cas immunity. Phage titer assay: top agar lawns supplemented with 5 mM CaCl$_2$ and inoculated with strains bearing the NHEJ system or not were poured on selective plates (with aTc for induction in *S. aureus*). We spotted serial dilutions of PhiNM4 or Phi2972 on the lawns of *S. aureus* and *S. thermophilus*, respectively. *Streptococcus aureus* strain RN4220 carried the *S. pyogenes* CRISPR-Cas system on plasmid pDB114 or a derivative with spacer 5′-AAAATGTTTTAACACCTATTAACGTAGTAT-3′ (pMD021). *S. thermophilus* strain DGCC7710 and a bacteriophage-insensitive mutant of strain DGCC7710 carrying spacer 5′-TGTTAAAAGAAGCACTA-GAGGTGATTTACG-3′ in the first position of the CRISPR-1 locus were used. EOP was determined after overnight incubation at 37 °C. Productive infection assays: cells were diluted 1:100 from overnight cultures in TSB supplemented with 5 mM CaCl$_2$ and the appropriate antibiotics and incubated at 37 °C. The NHEJ system was induced using aTc at OD600 0.2. After 30 min of incubation allowing the expression of the NHEJ system, we added phage PhiNM4 at an MOI (multiplicity of infection) of 1. Adsorption was allowed for 5 min at 37 °C with shaking. Cells were then put on ice and washed twice with ice-cold TSB. We then diluted and spotted them on top agar lawns of RN4220 supplemented with CaCl$_2$. ECOI was determined after overnight incubation at 37 °C.

**CRISPR-Cas interference efficiency assay using plasmids**. Cells carrying a type II-A CRISPR-Cas systems (pRH87) and the NHEJ system (pAB1) or the empty vector as a control (pE194) were made electro-competent as follows: cells were grown until OD 0.4, induced by adding aTc and further grown to OD 0.8. Cells were then washed twice with ice-cold water, once with 10% glycerol and resuspended in 1/100 of their volume in 10% glycerol. One hundred nanograms of plasmid pT181 or pAB2 were electroporated in 50 μL of competent cells (2500 V, 25 μF, 100 Ω, and 2 mm cuvettes). Cells were then incubated in 1 mL TSB for 1 h at 37 °C and plated on tetracycline only. Transformation efficiency was assessed after overnight incubation at 37 °C.

**Adaptation assays**. The spacer acquisition assay described below was adapted from ref. [10]. We mixed cells from overnight cultures (induced or non-induced) with phage (MOI value of 1) in top agar supplemented with 5 mM CaCl$_2$ and poured them on plates containing appropriate antibiotics and supplemented with aTc when necessary, followed by overnight incubation at 37 °C. For *S. aureus*, single colonies were resuspended in lysis buffer (250 mM KCl, 5 mM MgCl$_2$, 50 mM Tris-HCl at pH 9.0, 0.5% Triton X-100) supplemented with 20ng mL$^{-1}$ lysostaphin and incubated at 37 °C for 10 min, then 98 °C for 10 min. Following centrifugation (11 000 g), 1 μL of the supernatant was used as template for DreamTaqPCR amplification with primers AB23 and AB24. We provide a list of 15 acquired spacers in Supplementary Table 4. For *S. thermophilus*, single colonies were resuspended in 10 μL of water, 1 μL of which was used as template for DreamTaqPCR amplification with primers AB103 and AB104. The PCR reactions were analyzed on 2% agarose gels. Adaptation rates were computed as the estimated number of clones that acquired a spacer divided by the estimated number of cells in the initial population.

**Irradiation assay**. The NHEJ repair assay described below was adapted from ref. [41], 100 μl of overnight cultures of *B. subtilis* were irradiated at 100 Gy (RS Xstrahl, 42 min, 250 kV, 12 mA, 30 cm from focal point). We plated 1:10 000 dilutions on appropriate antibiotics. Colony-forming units (CFUs) were determined after overnight incubation at 37 °C. Survival rates were determined as the ratios of CFUs obtained for irradiated cells over CFUs obtained for non-irradiated cells.

**RNA extraction**. RNA was extracted from strains *B. subtilis* 168, *B. subtilis* 168 + pRH87, *B. subtilis* 168 + pAB56, *S. pyogenes* SF370, *S. aureus* RN4220 + pAB82, and *S. aureus* RN4220 + pAB1 + pRH87. Overnight cultures were diluted 1:100 in 2 mL and incubated at 37 °C for 3 h. For strains with pAB1 or pAB82 plasmids, aTc (0.5 μg μL$^{-1}$) was added after 1 h of incubation. Four milliliters of RNAprotect bacteria reagent (Qiagen) were added to the cultures, which were then vortexed briefly and incubated at room temperature for 5 min. The tubes were centrifuged at 4000×g for 5 min. Cell pellets of *B. subtilis* and *S. pyogenes* were resuspended in 200 μL of lysozyme buffer (lysozyme 20 mg mL$^{-1}$). *Streptococcus aureus* cell pellets were resuspended in 200 μL of lysostaphin solution (lysostaphin 5 mg mL$^{-1}$). After 1 h incubation at 37 °C, 1 mL of Trizol was added, and regular Trizol reagent procedures for purifying the total RNA were followed.

**RT-qPCR**. All the RNA samples were treated with DNase (Turbo DNase free kit, Ambion), then all the RNA samples (1 μg for each sample) were reverse transcribed into cDNA using the Transcriptor First strand cDNA synthesis Kit (Roche). The qPCR was performed using 1 μL of the reverse transcription reaction and the Faststart essential DNA green master mix (Roche) in a LightCycle 96 (Roche). Probes and PCR primers are listed in Supplementary Table 5. Relative gene expression was computed using the ΔΔCq method ($2^{Cq_{TAR}-Cq_{REF}}$), where $Cq_{REF}$ is the quantification cycle value for the 16s rRNA and $Cq_{TAR}$ for the tested gene. Data are shown relative to expression in the wild-type strain (Ku in *B. subtilis* 168 or Csn2 in *S. pyogenes* SF370).

**Data availability**. The datasets generated and analyzed during the current study are available from the corresponding author on reasonable request. Data corresponding to 5536 complete genomes retrieved from NCBI RefSeq information can be found in Supplementary Data 1.

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

## Acknowledgements

We thank Philippe Horvath for providing *S. thermophillus* strain DGCC7710 and phage Phi2972 and Sylvain Moineau for providing plasmid pNZ123. We also want to thank them both for their very helpful advice. We thank Matt Deyell for plasmid pMD021. A.B. is a member of the "Ecole Doctorale Frontière du vivant (FdV)". This work was supported by the European Research Council (ERC) under the Europe Union's Horizon 2020 research and innovation programme (grant CRISPAIR agreement no. 677823, and grant EVOMOBILOME no. 281605); the French Government's Investissement d'Avenir program; Laboratoire d'Excellence 'Integrative Biology of Emerging Infectious Diseases' (ANR-10-LABX-62-IBEID); the Pasteur-Weizmann consortium.

## Author contributions

A.B., E.P.C.R., M.T., and D.B. designed the research. A.B., A.C.-V., C.B., and L.C. performed the experiments. A.B., E.P.C.R., M.T., and D.B. wrote the manuscript.

## Additional information

**Competing interests:** The authors declare no competing financial interests.

