## [Peer Review File · Nature Communications]

Reviewers' comments:

Reviewer #1 (Remarks to the Author):

General comments.

The authors follow up their striking genomics-based observation suggesting an incompatibility between type II-A CRISPR-Cas systems and NHEJ in bacterial genomes by performing detailed functional analysis of CRISPR-Cas in the presence of NHEJ to show that NHEJ does not affect CRISPR-Cas, but rather the opposite that type II-A CRISPR-Cas interferes with DNA repair via NHEJ. This is a novel and, to me at least, fascinating finding, and the experimental work appears very well-executed. However, when discussing two systems that compete for the same substrates (dsDNA breaks), one should keep in mind that what system "wins" may be highly dependent on expression level of the competing systems. This is especially true in the context of heterologous expression of systems in a model host. I am not suggesting that all experiments be repeated switching the promoters of NHEJ and CRISPR-Cas, but some quantification of gene expression via Q-PCR or Northern would go a long way in convincing the reader that these phenotypes are real and biologically relevant.

The discussion should also elaborate on this limitation a little. It is obviously difficult to ascertain what would happen if an organism with NHEJ suddenly obtained a *csn2*. but the authors should at least inform the reader know whether *csn2*-based acquisition is tightly repressed or not in bacteria that naturally have it.

Another interesting point for the discussion is whether *Eggerthella* sp. YY7918 has homologs of recently discovered Anti-CRISPRs

Specific comments

Abstract

The first sentence is problematic, since the current understanding of type I CRISPR-Cas systems is that they do not generate double strand breaks, but rather nicks or single-stranded gaps, and that second-strand degradation may be due to other nucleases in the cell. Maybe revise to Type II Crispr-Cas systems or Class 2 CRISPR-Cas systems.

Line 35 "Yet even if" should be "Yet, although"

Line 38 "remains poorly understood" should be "remain"

Line 39 "in six types" should be "into six types"

Line 85 "suggesting strong negative epistasis." - I think "epistasis" is not the best term to use here, maybe consider "interaction"

Line 98 "there abundance co-vary with genome size" - should be "their abundance...".

Line 152-153 "efficiency of plaquing " should be "efficiency of plaquing (E.O.P)"

Line 298 "in agreement with our results..." - the argument made here is unclear, if NHEJ is weak then it should not affect the results of invasion assays (which is indeed the case), the sentence should be rephrased.

Line 317 "Hypothesis" should be "Hypotheses", since several are dicussed.

Line 319 "of a bacteria" - should be "of specific bacteria", or "a bacterium"

Line 342 "epistatic interactions" - here the "epistatic" is probably doing more harm than good, as there maybe physical interactions in some cases, not just genetic ones.

The Materials and Methods section has multiple grammatical errors that should be fixed, but

Nature Comms. probably provides language editing, so I saved myself the trouble.

Reviewer #2 (Remarks to the Author):

In this study, Bernheim et al., examine the genetic and functional overlap between NHEJ and the Type II-A CRISPR-CAA pathways in bacteria. Specifically, they show using in silico analysis that there is a negative association between the NHEJ and Type II-A CRISPR systems in bacteria, particularly in Firmicutes. They go on to show experimentally that NHEJ does not interfere with CRISPR interference in *S.aureus* cells and it also does not interfere with spacer acquisition in vivo. Finally, they present data that suggests that Csn2 may block NHEJ by competing for Ku binding to the ends of DSBs.

This report reads more like a project report or supporting supplementary data to a more major story than an actual study on its own. The data presented are very preliminary, mainly negative and the positive data are relatively unremarkable or surprising given Ku and Csn2 bind to similar substrates. For example, if you overexpress a range of end-binding factors you'd see the same result but this does not prove that they will be found to be mutually exclusive in nature.

In summary, this is a relatively preliminary and incremental study and that lacks substantial experimental data and whose findings are more suitable as a brief communication to a specialized journal, but even then only after more significant experimental work has been included.

Reviewer #1

Comment 1.1. The authors follow up their striking genomics-based observation suggesting an incompatibility between type II-A CRISPR-Cas systems and NHEJ in bacterial genomes by performing detailed functional analysis of CRISPR-Cas in the presence of NHEJ to show that NHEJ does not affect CRISPR-Cas, but rather the opposite that type II-A CRISPR-Cas interferes with DNA repair via NHEJ. This is a novel and, to me at least, fascinating finding, and the experimental work appears very well-executed. However, when discussing two systems that compete for the same substrates (dsDNA breaks), one should keep in mind that what system "wins" may be highly dependent on expression level of the competing systems. This is especially true in the context of heterologous expression of systems in a model host. I am not suggesting that all experiments be repeated switching the promoters of NHEJ and CRISPR-Cas, but some quantification of gene expression via q-PCR or Northern would go a long way in convincing the reader that these phenotypes are real and biologically relevant.

Answer. Given the peculiarities of the work (two systems that never co-occur), there are no models where we can use the native expression systems in the native background for the two systems simultaneously. Nevertheless, we agree with the comment that expression levels are necessarily going to affect the outcome of the experience and that over-expression of *csn2* could result in biologically irrelevant results (especially if NHEJ is simultaneously under-expressed). We performed qPCR measurements to assess the problem of potential overexpression of the exogenous system in *S. aureus* and *B. subtilis*. Our results show that *csn2* in *B. subtilis* was slightly under-expressed relative to the basal level in the native host, whereas NHEJ was over-expressed in *S. aureus* relative to the level found in the native host. Hence, expression data suggests that, if anything, we are under-estimating the effect of the competition.

Action. We added to the manuscript Supplementary Text 2, Supplementary Figure 6 and the following lines in the main text.

In the part about NHEJ inhibition in *B. subtilis*:

“In this set of experiments a possible concern is that Csn2 might be overexpressed which could lead to artifacts with no biological relevance. To prevent this issue, we expressed the whole *S. pyogenes* type II-A system or Csn2 alone from the natural promoter of the *cas* operon (plasmid pRH87 and pAB56 respectively). The expression of Csn2 in *B. subtilis* as measured by qPCR was 3.6-fold lower than the basal expression level of Csn2 in *S. pyogenes* SF370 (Supplementary Text 2 and Supplementary Figure 6). This low level of expression might reflect what would happen after a natural horizontal gene transfer event.”

And then in the part about NHEJ inhibition in *S. aureus*:

“In this assay the NHEJ system is strongly overexpressed compared to the natural expression of Ku and LigD in *B. subtilis* during stationary phase. Note that such overexpression was necessary to observe plasmid recircularization events in *S. aureus*. On the other hand, Csn2 was only slightly overexpressed compared to its expression level in *S. pyogenes* SF370.”

Supplementary Figure 6 : Expression level of NHEJ and Csn2

Expression of NHEJ and Csn2 was measured using q-PCR in strains used in the study (see Supplementary Table 3 for a description of the plasmids). Expression was normalized to the 16s rRNA expression in each strain measured using the same set of primers. Expression is shown relative to the wild-type: *B. subtilis* 168 for NHEJ and *S. pyogenes* SF370 for Csn2.

Supplementary Text 2: Expression of NHEJ and Csn2 in strains used in the study

RNA extraction

RNA was extracted from strains *B. subtilis* 168, *B. subtilis* 168 + pRH87, *B. subtilis* 168 + pAB56, *S. pyogenes* SF370, *S. aureus* RN4220 + pAB82, *S. aureus* RN4220 + pAB1 + pRH87. Overnight cultures were diluted 1:100 in 2ml and incubated at 37°C for 3 hours. For strains with pAB1 or pAB82 plasmids, aTc (0.5 ug/ul) was added after 1 hour of incubation. 4 ml of RNeasy Protect Bacteria reagent (Qiagen) were added to the cultures, which were then vortexed briefly and incubated at room temperature for 5 minutes. The tubes were spun down at 4000 g for 5 minutes. Cell pellets of *B. subtilis* and *S. pyogenes* were resuspended in 200 ul of lysis buffer (lysozyme 20 mg/ml). *S. aureus* cell pellets were resuspended in 200ul of lysis solution (lysostaphin 5mg/mL). After 1 hour incubation at 37 °C, 1 ml of trizol was added, and regular trizol reagent procedures for purifying the total RNA were followed.

RT-qPCR

All the RNA samples were treated with DNase (Turbo DNase free kit, Ambion), then all the RNA samples (1 ug for each sample) were reverse transcribed into cDNA using the Transcriptor First strand cDNA synthesis Kit (Roche). The qPCR was performed using 1 ul of the reverse transcription reaction and the Faststart essential DNA green master mix (Roche) in a LightCycle 96 (Roche). Probes and PCR primers are listed below. Relative gene expression was computed using the $\Delta\Delta C_q$ method ($2^{C_{q_TAR} - C_{q_REF}}$) where C_{q_REF} is the quantification cycle value for the 16s rRNA and C_{q_TAR} for the tested gene. Data is shown relative to expression in the wild-type strain (Ku in *B. subtilis* 168 or Csn2 in *S. pyogenes* SF370).

Targeted genes	Primer name	Sequences (5' to 3')
Ku	LC1340_Ku_For	GGATCGATCAGCTTCGGATTAG
Ku	LC1341_Ku_Rev	TGGTGCGTGATCCTCTTTATG
Csn2	LC1342_csn2_For	GCAAACCTCCGATGAAAGACTTG
Csn2	LC1343_csn2_Rev	ACCGCCTCTTAATGGAATCG
16s_rRNA	LC1344_16s_For	AGGCAGCAGTAGGGAATCTT
16s_rRNA	LC1345_16s_Rev	GCTGCTGGCACGTAGTTAG

Comment 1.2. The discussion should also elaborate on this limitation a little. It is obviously difficult to ascertain what would happen if an organism with NHEJ suddenly obtained a *csn2*, but the authors should at least inform the reader whether *csn2*-based acquisition is tightly repressed or not in bacteria that naturally have it.

Answer. In our opinion the most likely conflict arises in a genome encoding NHEJ and receiving a CRISPR-Cas system by HGT (because the latter seem to be much more transferred than the former). This fits the scenario of the reviewer. We have shown above (answer #1.1) that *csn2* is actually less expressed in the novel host than in the native host, which is the most likely case for HGT genes.

Action. The expression data of *Csn2* in *S. pyogenes* is indicated in Supplementary figure 6. We also added the following sentence in the discussion:

“Note that type II-A systems are constitutively expressed in the bacteria where they have been studied (*S. pyogenes*⁷, *S. thermophilus*⁴⁶), and would thus likely also be expressed in the recipient upon horizontal gene transfer.”

Comment 1.3. Another interesting point for the discussion is whether *Eggerthella* sp. YY7918 has homologs of recently discovered Anti-CRISPRs.

Answer. We used blastp to detect potential sequences of known anti-CRISPRs against type II systems as they were described in ¹⁻³. We found no homologs in *Eggerthella* sp YY918. However, many more anti CRISPRs remain to be discovered and one cannot exclude the possibility that *Eggerthella* sp YY918 harbours an anti-CRISPR.

Action. We added the following underlined sentence to the main text:

“Only one genome among the 5563 encodes both NHEJ and type II-A: the actinobacteria *Eggerthella* sp. YY7918. In this genome, both NHEJ and type II-A systems seem intact, since the *cas* operon contains all four genes, lacking frameshifts or premature stop codons, and the adjacent CRISPR array encodes 44 spacers. We were also unable to detect anti-CRISPR proteins similar to the ones described in the literature³⁴⁻³⁶.”

Specific comments

Abstract

The first sentence is problematic, since the current understanding of type I CRISPR-Cas systems is that they do not generate double strand breaks, but rather nicks or single-stranded gaps, and that second-strand degradation may be due to other nucleases in the cell. Maybe revise to Type II Crispr-Cas systems or Class 2 CRISPR-Cas systems.

The abstract was modified as follow:

“Type II CRISPR-Cas systems introduce double strand breaks into DNA of invading genetic material”

We would like to thank the reviewer for his careful proof-reading. All the points below were corrected in the manuscript as suggested.

Line 35 "Yet even if" should be "Yet, although"

Line 38 "remains poorly understood" should be "remain"

Line 39 "in six types" should be "into six types"

Line 85 "suggesting strong negative epistasis." - I think "epistasis" is not the best term to use here, maybe consider "interaction"

Line 98 "there abundance co-vary with genome size" - should be "their abundance...".

Line 152-153 "efficiency of plaquing" should be "efficiency of plaquing (E.O.P)"

Line 317 "Hypothesis" should be "Hypotheses", since several are dicussed.

Line 319 "of a bacteria" - should be "of specific bacteria", or "a bacterium"

Line 342 "epistatic interactions" - here the "epistatic" is probably doing more harm than good, as there maybe physical interactions in some cases, not just genetic ones.

Line 298 "in agreement with our results..." - the argument made here is unclear, if NHEJ is weak then it should not affect the results of invasion assays (which is indeed the case), the sentence should be rephrased.

Our sentence was indeed misleading, we reformulated it as follow:

“Previous studies showed that NHEJ repair pathways are able to repair Cas9-mediated DNA breaks in various bacterial species^{19,37}. The efficiency of repair in these experimental setups was very low. Consistently, our results show that NHEJ repair cannot lead to a meaningful reduction in phage infectivity or plasmid transfer.”

The Materials and Methods section has multiple grammatical errors that should be fixed, but Nature Comms. probably provides language editing, so I saved myself the trouble.

We carefully corrected the Methods section and indeed caught several grammatical errors.

Reviewer #2 :

Comment 2.1. In this study, Bernheim et al., examine the genetic and functional overlap between NHEJ and the Type II-A CRISP-CAA pathways in bacteria. Specifically, they show using in silico analysis that there is a negative association between the NHEJ and Type II-A CRISPR systems in bacteria, particularly in Firmicutes. They go on to show experimentally that NHEJ does not interfere with CRISPR interference in *S. aureus* cells and it also does not interfere with spacer acquisition in vivo. Finally, they present data that suggests that Csn2 may block NHEJ by competing for Ku binding to the ends of DSBs.

This report reads more like a project report or supporting supplementary data to a more major story than an actual study on its own. The data presented are very preliminary, mainly negative and the positive data are relatively unremarkable or surprising given Ku and Csn2 bind to similar substrates. For example, if you overexpress a range of end-binding factors you'd see the same result but this does not prove that they will be found to be mutually exclusive in nature.

In summary, this is a relatively preliminary and incremental study and that lacks substantial experimental data and whose findings are more suitable as a brief communication to a specialized journal, but even then only after more significant experimental work has been included.

Answer. There are three criticisms here. We answer them separately.

Preliminary work. Our manuscript includes data gathered from 3 different model bacteria, all of them including novel genetic and experimental setups to measure CRISPR interference (through phage infection in *S. aureus* and *S. thermophilus*; and plasmid transfer assay in *S. aureus*), CRISPR adaptation (in *S. aureus* and in *S. thermophilus*) and NHEJ activity (in *B. subtilis* and in *S. aureus*), all this together with a careful bioinformatics study using thousands of genomes in which system interaction was controlled by phylogeny and that would be worthy of publication in itself. Hence, we cannot agree with the dismissive comments of the reviewer that this corresponds to a very preliminary study. We don't know of many studies in the literature with significant computational and experimental work while including the genetic manipulation of three very distinct model systems.

Negative results. We included in our manuscript some negative results because this is essential to separate the different possible explanations for the computational results. We approached the surprising bioinformatics observations that NHEJ and type II-A CRISPR-Cas systems avoid each other in an unbiased way and tested all the possible hypothesis experimentally. Not publishing the negative results would leave the question open if expressing Ku and LigD could have a strong impact on CRISPR interference and immunization. While our results do not enable us to completely exclude the hypothesis that NHEJ proteins could affect CRISPR immunity, we observed no effects in experimental setups where Ku and LigD were strongly overexpressed. In a general context where many voices are promoting the publication of negative results, we feel like this manuscript is a more than adequate avenue to share these data, especially considering the fact that they come alongside positive results regarding the effect of Csn2 on NHEJ repair.

Expression and interference. The reviewer assumes that the results are caused by the overexpression of *csn2* genes in our experimental setups. The importance of assessing gene expression was also made by reviewer 1 (see answers to comments #1.1 and #1.2). To address these concerns, we performed qPCR measurements and showed that *csn2* is not over-expressed at

all (on the contrary, it is less expressed in *B. subtilis* than in the native background). Hence, the criticism that the key result is based on the over-expression of *csn2* is factually incorrect.

Action. We added Supplementary Text 2, Supplementary Figure 6 (pasted above) and the following lines to the manuscript.

In the part about NHEJ inhibition in *B. subtilis*:

“In this set of experiments a possible concern is that Csn2 might be overexpressed which could lead to artifacts with no biological relevance. To prevent this issue, we expressed the whole *S. pyogenes* type II-A system or Csn2 alone from the natural promoter of the cas operon (plasmid pRH87 and pAB56 respectively). The expression of Csn2 in *B. subtilis* as measured by qPCR was 3.6-fold lower than the basal expression level of Csn2 in *S. pyogenes* SF370 (Supplementary Text 2 and Supplementary Figure 6). This low level of expression might reflect what would happen after a natural horizontal gene transfer event.”

And then in the part about NHEJ inhibition in *S. aureus*:

“In this assay the NHEJ system is strongly overexpressed compared to the natural expression of Ku and LigD in *B. subtilis* during stationary phase. Note that such overexpression was necessary to observe plasmid recircularization events in *S. aureus*. On the other hand, Csn2 was only slightly overexpressed compared to its expression level in *S. pyogenes* SF370.”

We would also like to point out that our data contradicts that reviewer’s claim that over-expressing any competing enzymes will lead to their cross inhibition. The overexpression of NHEJ genes did not affect spacer acquisition or interference in *S. aureus* nor in *S. thermophilus*.

REVIEWERS' COMMENTS:

Reviewer #1 (Remarks to the Author):

The authors have completely and convincingly addressed my concerns in their revised manuscript. I feel that the manuscript makes a unique contribution to both CRISPR and DNA repair fields, and I am confident that it will generate much interest in even broader contexts.